# β2-integrins as biomarkers in urothelial cancer

**Marc Llort Asens**, **Imran Khan**, **Susanna Carola Fagerholm**\*

MIBS research programme, Faculty of Biological and Environmental Sciences, University of Helsinki, Helsinki, Finland

\* susanna.fagerholm@helsinki.fi

## Abstract

β2-integrins are a family of adhesion proteins expressed in immune cells that play multiple roles in anti-tumor immunity. β2-integrins regulate tumor infiltration of anti-tumorigenic immune cells such as cytotoxic CD8+T cells and NK cells. However, they also regulate the activity of myeloid cells, such as macrophages, which can have both anti- and pro-tumorigenic properties. The role of β2-integrins in urothelial cancer remains poorly understood. Here, we have investigated the role of different β2-integrins, and their cytoplasmic regulators, in urothelial cancer, by utilizing RNA expression data. We found that *ITGAL* (encoding for CD11a) and *FERMT3* (encoding for the integrin regulator kindlin-3) have a positive correlation with patient survival. EcoTyper analysis revealed increased infiltration of CD8+T cells and NK cells in *ITGAL* high samples, but *ITGAL* or *FERMT3* expression did not correlate with response to immunotherapy. In contrast, *ITGAM* and *ITGAX* (which encode for myeloid markers CD11b and CD11c) and *FLNA* (encoding for the integrin regulator filamin A) correlated with poor survival and reduced responsiveness to immunotherapy and critically regulate the tumor myeloid immune landscape (M1/M2 macrophages, cDC1 dendritic cells). Therefore, β2-integrins may be explored in the future as biomarkers to differentiate urothelial cancer patients with different immune landscapes, responding differently to therapy.

## Introduction

Cancer remains a leading global health challenge, nearly 20 million people worldwide were diagnosed with cancer in 2020. Urothelial bladder cancer was ranked 9th in incidence and 13th in cancer related mortality as per World Health Organization, accounting to over 614,000 cases and more than 220,000 deaths [1] As for other cancer types, immunotherapy is used also in urothelial bladder cancer, but response rates lie below 30% [2,3]. This highlights an urgent need for better understanding of molecular mechanisms behind urothelial bladder cancer development, characterization of the immune landscape of the disease, and identification of reliable novel biomarkers predicting disease progression and response to therapy [4].

**Data availability statement:** The data underlying the results presented in the study are available from: IMvigor210 urothelial cancer cohort http://research-pub.gene.com/IMvigor210CoreBiologies/ The original data underlying the results are available from: https://ega-archive.org/studies/EGAS00001002556.

**Funding:** Research Council of Finland Liv och Hälsa foundation Magnus Ehrnrooth Foundation The funders had no role in study design, data collection and analysis, decision to publish, or preparation of the manuscript.

**Competing interests:** The authors have declared that no competing interests exist.

β2-integrins are a family of transmembrane heterodimeric proteins located at the surface of all leukocytes [5]. They play an essential role in immune cell trafficking to inflammation sites and tumors by allowing leukocytes to move between the blood stream and the surrounding tissues [6]. Among many other functions, they are also key to the modulation of the immune system, by regulating the activation state and polarization of myeloid cells such as macrophages and dendritic cells. β2-integrins are also central in the formation of immunological synapses, complex structures that immune cells form to communicate between themselves and to mediate tumor cell killing [6].

Despite sharing a common β subunit, encoded by the gene *ITGB2*, β2-integrins differ in their α subunits and their expression can be restricted to different cell types [5]. Thus, the α subunit encoded by the gene *ITGAL* pairs with β2 to form the integrin αLβ2 (also known as LFA-1), which is expressed in all leukocytes and is the predominant β2-integrin in lymphocytes. The α subunit encoded by the gene *ITGAM* forms the integrin αMβ2 (also called Mac-1) and is mostly expressed in myeloid cells (particularly in neutrophils, but also in macrophages and dendritic cells). The α subunit encoded by the gene *ITGAX* forms the integrin αXβ2 (CD11c/CD18), which is most abundant in dendritic cells. Lastly, the α subunit encoded by the gene *ITGAD* encodes for the integrin αDβ2, the most recently discovered and least known β2-integrin, which is expressed in neutrophils, monocytes and natural killer cells.

Integrins are regulated by interactions with cytoplasmic proteins binding to their cytoplasmic domains (tails). For example, talin (encoded by *TLN1*) and kindlin-3 (encoded by *FERMT3*) are essential for integrin activation and binding to ligands [7–11], whilst filamin A (encoded by *FLNA*) has been reported to play both positive and negative roles in integrin function [12,13]

Due to their heterogeneity in both structure and expression profile, it is not surprising that different β2-integrins perform different functions, at times even completely opposite roles in the modulation of the immune system both in health and in disease. Hence, while αLβ2 is key to trafficking of cytotoxic CD8 + T cells into tumors and CD8 + T cell effector functions [6], αMβ2 can play an opposite role in myeloid cells such as macrophages and dendritic cells (DCs) by suppressing their activation, function as antigen presenting cells, activators of T cells and anti-tumor responses [14–17]. However, the role of β2-integrins in regulating urothelial bladder cancer immune landscape and/or responsiveness to therapy remains poorly understood.

Here, we have investigated the role of β2-integrins and their cytoplasmic regulators in urothelial cancer, by utilizing the IMvigor210 cohort (NCT02108652). We found that although expression of *ITGAL* and *FERMT3* (which regulate immune cell infiltration into tumors) do correlate with increased survival, they do not predict response to immunotherapy. In contrast, myeloid markers *ITGAM* and *ITGAX* were inversely correlated with patient survival and response to immunotherapy, and also critically regulate the myeloid immune landscape of the tumors. Therefore, different subclasses of β2-integrins may be used as biomarkers to predict disease outcome and responsiveness to immunotherapy in urothelial cancer.

## Results

### T cell markers expression levels effect on survival and response to immunotherapy

We investigated whether expression levels of *ITGB2*, *ITGAL*, and *FERMT3*, which are important for leukocyte (T cell, NK cell) infiltration into tumors, had an effect on cancer patient survival. To that aim, we used the publicly available gene expression data from the IMvigor210 study, which evaluates the effect of anti-PD-L1 immunotherapy on survival of patients of urothelial cancer. We categorized the patients into two separate groups (high or low) based on individual gene expression levels of *ITGB2*, *ITGAL*, and *FERMT3* (see Materials and Methods). We observed no significant differences in the length of overall survival between patients with high or low expression of *ITGB2*, which encodes for the common β2-chain of all four β2-integrins ([Fig 1A]). In contrast, we identified that patients with a high expression of *ITGAL* (*ITGAL* high group) showed a significantly longer overall survival than the patients with low *ITGAL* expression (*ITGAL*

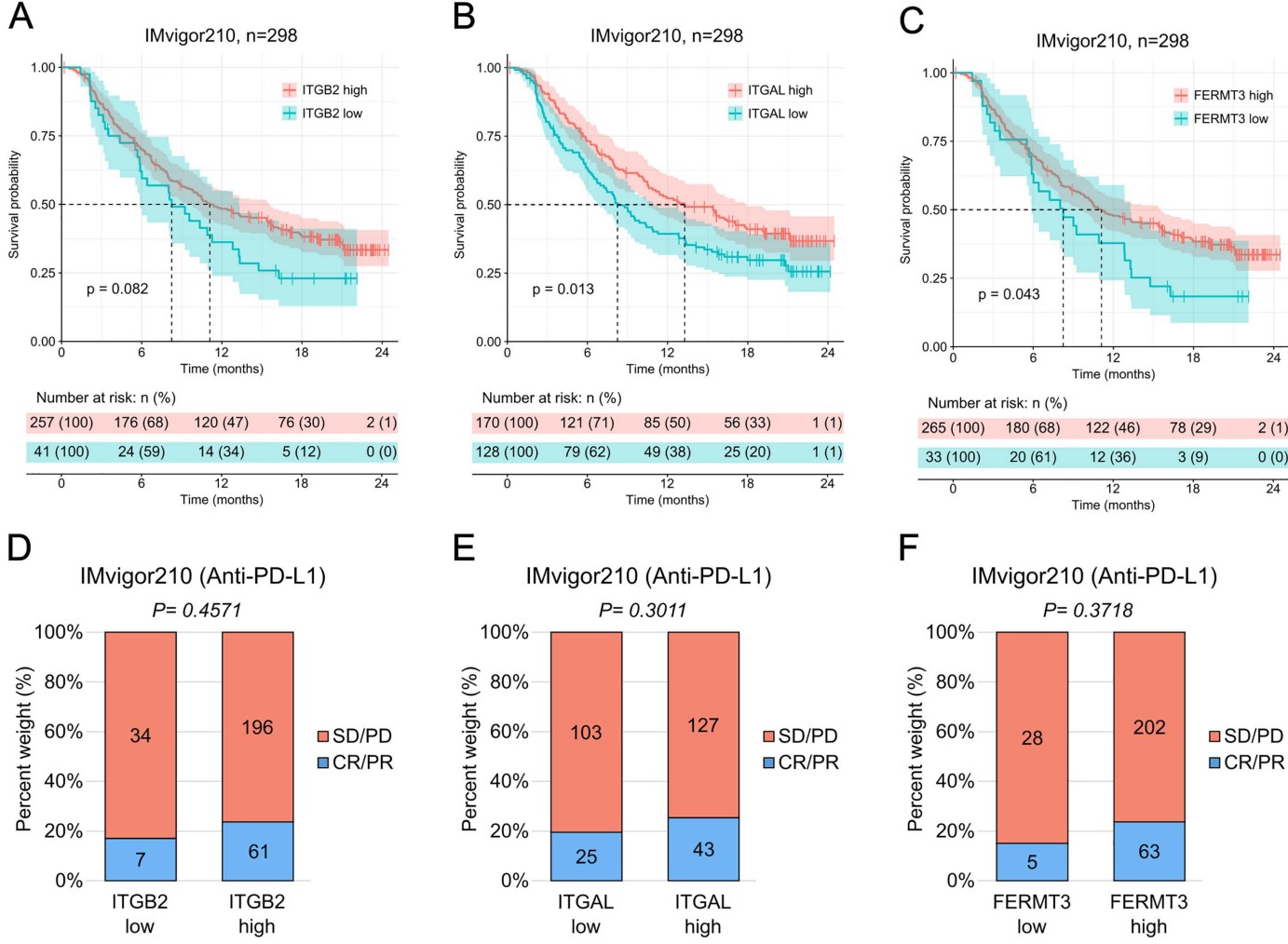

**Fig 1. High expression levels of *ITGAL* and *FERMT3* correlate with longer overall survival in urothelial cancer patients. A-C**, effect of gene expression in infiltrating leukocytes. Kaplan-Meier estimator of the overall survival after treatment with anti-PD-L1 in the IMvigor210 cohort of *ITGB2* high (n = 257) and *ITGB2* low (n = 41) patients **(A)**, *ITGAL* high (n = 170) and *ITGAL* low (n = 128) patients **(B)**, and *FERMT3* high (n = 265) and *FERMT3* low (n = 33) patients **(C)**. The dotted lines indicate the time at which each group reached median survival (*ITGB2* high: 11.1 months, *ITGB2* low: 8.25 months; *ITGAL* high: 13.27 months, *ITGAL* low: 8.25 months; *FERMT3* high: 11.1 months, *FERMT3* low: 8.25 months).

low) [P = 0.013] and that their group reached the median of survival probability remarkably later (5 months) than the group with low *ITGAL* expression (Fig 1B). In line with these findings, we also identified that patients with a high expression of *FERMT3* (*FERMT3* high) showed a significantly longer overall survival than the patients with low *FERMT3* expression (*FERMT3* low) [P = 0.043] (Fig 1C).

However, the expression levels of *ITGB2*, *ITGAL* or *FERMT3* did not significantly affect how well the patients responded to anti-PD-L1 immunotherapy as accounted by the number of patients with stable disease or progressive disease (SD/PD) versus the number of patients with a complete response or partial response (CR/PR) (Fig 1D–F).

Together, these results highlight the importance of *ITGAL* and *FERMT3* in the immune response against urothelial cancer.

The colored vertical marks on the plot indicate censored events for each group in time. Below the plot, risk table indicating the number of patients at risk belonging to each group at each point in time: high, in orange, or low, in blue. Statistical analysis was done using the log-rank (Mantel-Cox) test. P-values are shown on the plots. **D-F**, rate of clinical response to anti-PD-L1 immunotherapy in the IMvigor210 cohort (SD/PD, stable disease/progressive disease, n = 257 (**D**), n = 170 (**E**), n = 265 (**F**); CR/PR, complete response/partial response, n = 41 (**D**), n = 128 (**E**), n = 33 (**F**)). Statistical analysis was done using the chi-squared test. P-values are shown on the plots.

### Myeloid β2-integrin expression levels impact on survival and immunotherapy response

Next, we wanted to study whether the expression levels of *ITGAM*, *ITGAX* and *ITGAD*, which are typically expressed in myeloid cells and are associated with suppression of the immune response, affected patient survival and response to anti-PD-L1 immunotherapy in the urothelial cancer cohort.

We found that patients with low levels of *ITGAM* (*ITGAM* low) or *ITGAX* (*ITGAX* low), analyzed independently, showed significantly longer overall survival compared to those with high levels of expression (*ITGAM* high, and *ITGAX* high, respectively) (Fig 2A–B). Additionally, they responded significantly better to anti-PD-L1 immunotherapy (Fig 2D–E).

In contrast, the expression levels of *ITGAD* did not seem to alter significantly either the overall survival (Fig 2C) or the response to anti-PD-L1 immunotherapy (Fig 2F) in patients with urothelial cancer.

Together, these results emphasize the importance of *ITGAM* and *ITGAX* expression levels in urothelial cancer progression and response to anti-PD-L1 immunotherapy.

### β2-integrin regulator expression levels effect on survival and response to immunotherapy

We further examined how the expression level of two-well known β2-integrin regulators, *FLNA* and *TLN1*, affected the overall survival and response to anti-PD-L1 immunotherapy in urothelial cancer.

Remarkably, patients with low levels of *FLNA* (*FLNA* low) had a significantly longer overall survival compared to those with high levels of *FLNA* (*FLNA* high) and reached median survival considerably later (almost 5-month delay) (Fig 3A). They also responded significantly better to anti-PD-L1 immunotherapy (Fig 3C). In contrast, different expression levels of *TLN1* did not seem to significantly alter the overall survival of the patients (Fig 3B) or their response to immunotherapy (Fig 3D).

Together, these results underline a key role for *FLNA* expression levels in urothelial cancer progression and response to anti-PD-L1 immunotherapy.

### Association of *ITGAL* and *FERMT3* expression levels with adaptive immune response genes

Our results indicated that different β2-integrins and their regulators have very different roles in urothelial cancer development. Given β2-integrins are immune-specific proteins, we sought to investigate their role in the immune landscape of the tumors. To this end, we performed differential gene expression analysis on each of the eight genes of interest, comparing the groups with low versus high expression levels.

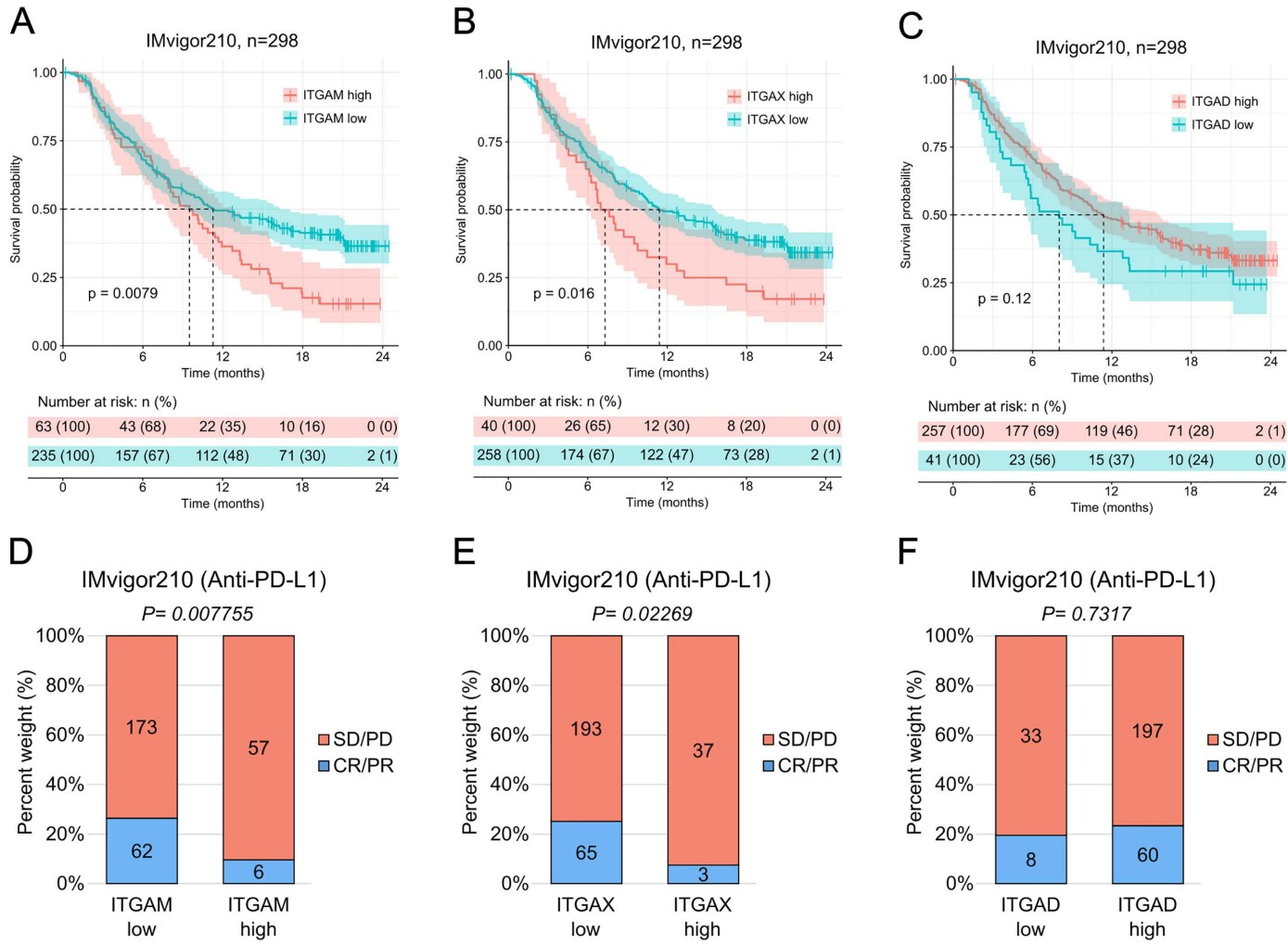

**Fig 2. Low expression levels of *ITGAM* and *ITGAX* correlate with longer overall survival in urothelial cancer patients and enhanced response to anti-PD-L1 immunotherapy. A-C**, Kaplan-Meier estimator of the overall survival after treatment with anti-PD-L1 of *ITGAM* high (n = 63) and *ITGAM* low (n = 235 patients **(A)**, *ITGAX* high (n = 40) and *ITGAX* low (n = 258) patients **(B)**, and *ITGAD* high (n = 257) and *ITGAD* low (n = 41) patients **(C)**. The dotted lines indicate the time at which each group reached median survival (*ITGAM* high: 9.49 months, *ITGAM* low: 11.27 months; *ITGAX* high: 7.29 months, *ITGAX* low: 11.4 months; *ITGAD* high: 11.36 months, *ITGAD* low: 8.02 months). The colored vertical marks on the plot indicate censored events for each group in time. Below the plot, risk table indicating the number of patients at risk belonging to each group at each point in time: high, in orange, or low, in blue. Statistical analysis was done using the log-rank (Mantel-Cox) test. P-values are shown on the plots. **D-F**, rate of clinical response to anti-PD-L1 immunotherapy (SD/PD, stable disease/progressive disease, n = 63 **(D)**, n = 40 **(E)**, n = 257 **(F)**; CR/PR, complete response/partial response, n = 235 **(D)**, n = 258 **(E)**, n = 41 **(F)**). Statistical analysis was done using the chi-squared test. P-values are shown on the plots.

We observed a significant correlation of β2-integrins and immune genes in the samples (Fig 4). For example, down-regulated markers in the *ITGAL* (Fig 4B) and *FERMT3* (Fig 4D) low group revealed pathways such as "leukocyte activation", "adaptive immune response", "positive regulation of immune response", "leukocyte migration", "humoral immune response", "extrafollicular and follicular B cell activation by SARS CoV2" and "regulation of cell killing". For instance, genes such as *CD2* (a costimulatory molecule involved in immunological synapse formation), ZAP70 (a tyrosine kinase which has an essential role in T cell activation), IL2RB (the IL-2 receptor which is essential for T cell function), BTLA (a lymphocyte co-signaling molecule of the CD28 superfamily), and KLRG1 (an inhibitory lectin-like receptor expressed on

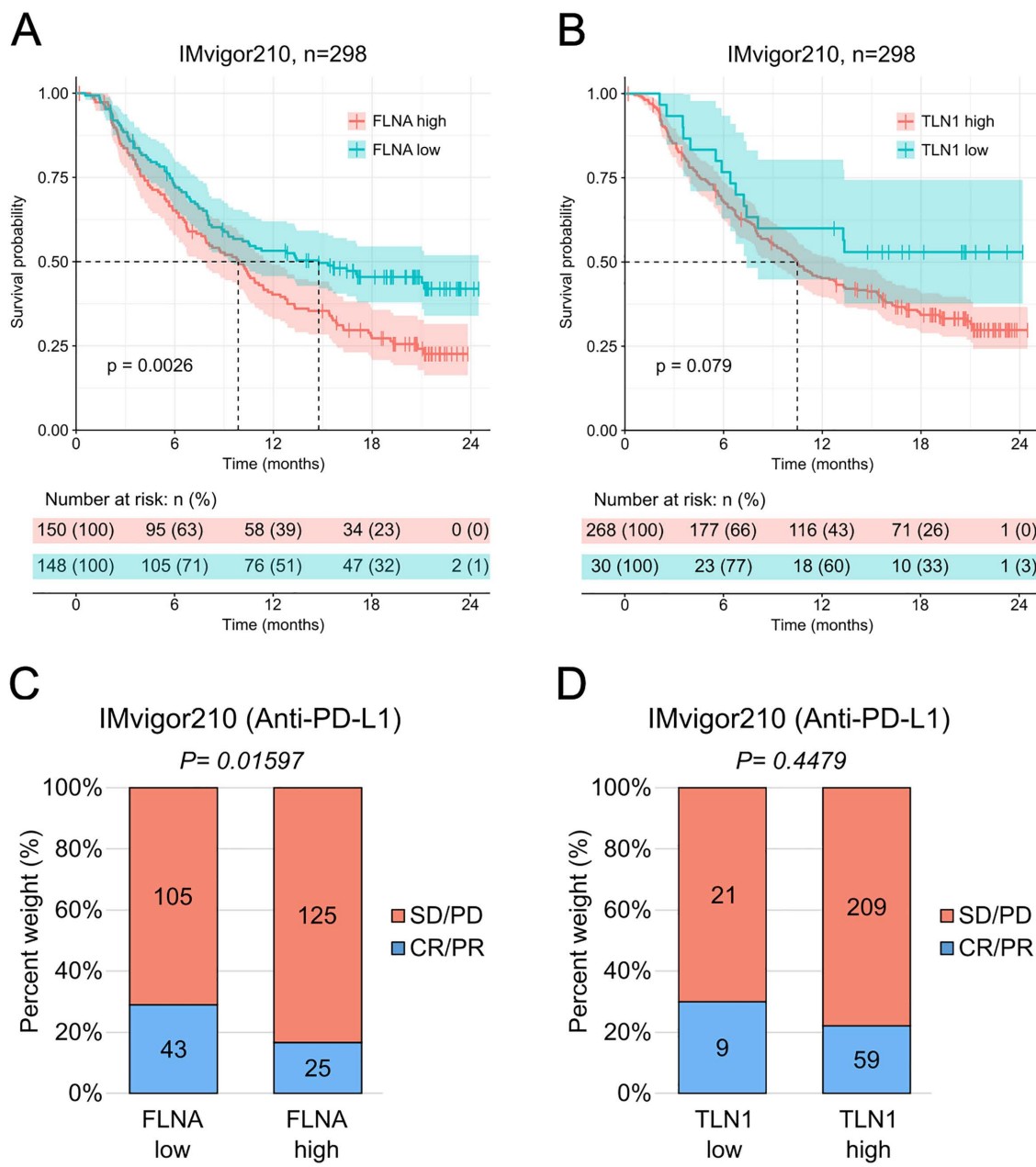

**Fig 3. Low expression levels of *FLNA* correlate with longer overall survival in urothelial cancer patients and enhanced response to anti-PD-L1 immunotherapy. A-B**, effect of gene expression in infiltrating leukocytes. Kaplan-Meier estimator of the overall survival after treatment with anti-PD-L1 of *FLNA* high (n = 150) and *FLNA* low (n = 148) patients **(A)**, and *TLN1* high (n = 268) and TLNA low (n = 30) patients **(B)**. The dotted lines indicate the time at which each group reached median survival (*FLNA* high: 9.86 months, *FLNA* low: 14.75 months; *TLN1* high: 10.48 months, *TLN1* low: not applicable). The colored vertical marks on the plot indicate censored events for each group in time. Below the plot, risk table indicating the number of patients at risk belonging to each group at each point in time: high, in orange, or low, in blue. Statistical analysis was done using the log-rank (Mantel-Cox) test. P-values are shown on the plots. **C-D**, rate of clinical response to anti-PD-L1 immunotherapy (SD/PD, stable disease/progressive disease, n = 150 **(C)**, n = 268 **(D)**; CR/PR, complete response/partial response, n = 148 **(C)**, n = 30 **(D)**). Statistical analysis was done using the chi-squared test. P-values are shown on the plots.

T cells and NK cells) were downregulated in *ITGAL* low group (Fig 4A). Low expression of *ITGAL* and *FERMT3* therefore appears associated with suppressed adaptive immune responses in the tumors.

## Association of *ITGAM* and *ITGAX* expression levels with myeloid immune response genes

In contrast to *ITGAL* and *FERMT3*, *ITGAM* and *ITGAX* showed negative association with survival, and also negative correlation with response to immunotherapy (Fig 2). The differential gene expression analysis again revealed association of these markers with immune genes (Fig 5B and 5D), but the pathways identified were in part different from *ITGAL/FERMT3*. For instance, in the *ITGAM* low group, pathways such as "inflammatory response", "neutrophil degranulation", "myeloid leukocyte activation", "innate immune response" and "phagosome" were downregulated compared to the *ITGAM* high group (Fig 5B). For example, OSCAR (a FcRγ associated receptor expressed in myeloid cells, involved in antigen presentation), CCL18 (an immunosuppressive, pro-tumorigenic chemokine produced by dendritic cells and macrophages, associated with M2 macrophage polarization), CHIT1 (an enzyme with important roles in macrophage biology), LTF and MPO (enzymes found in myeloid cells such as neutrophils) were downregulated in the *ITGAM/ITGAX* low groups (Fig 5A and 5C). This analysis clearly revealed an association of *ITGAM* (and in part *ITGAX*) with the myeloid immune landscape of the tumors.

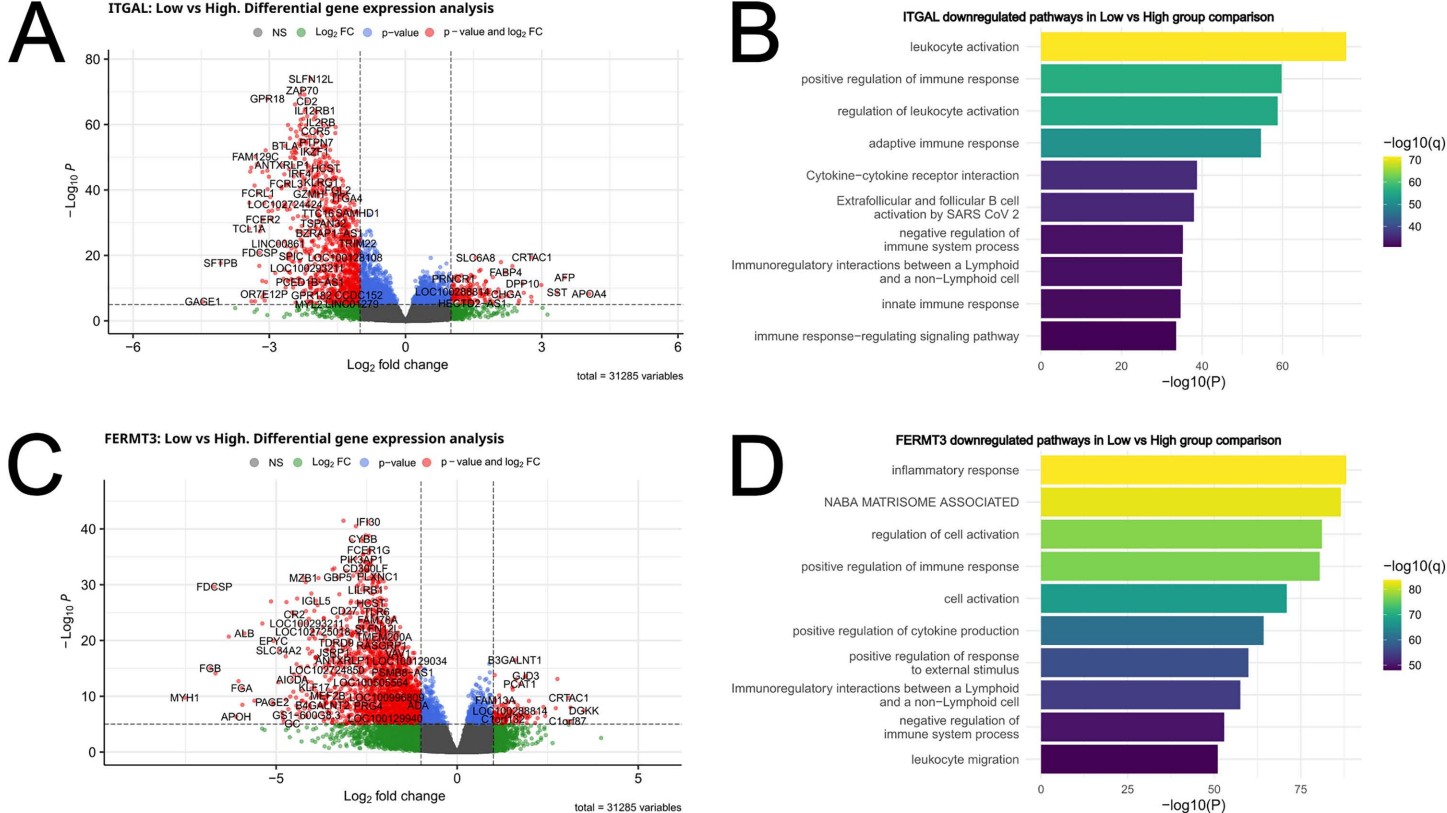

**Fig 4. Differential gene expression analysis and gene set enrichment analysis for *ITGAL* and *FERMT3*.** Volcano plots showing the differentially expressed genes in the group comparison low vs high for *ITGAL* (A) and *FERMT3* (C). All genes that passed the p-value and log2 fold change threshold are indicated in red. Downregulated genes in the comparison have a negative Log2 fold change and upregulated genes have a positive Log2 fold change. Functionally enriched pathways were plotted for *ITGAL* (B) and *FERMT3* (D).

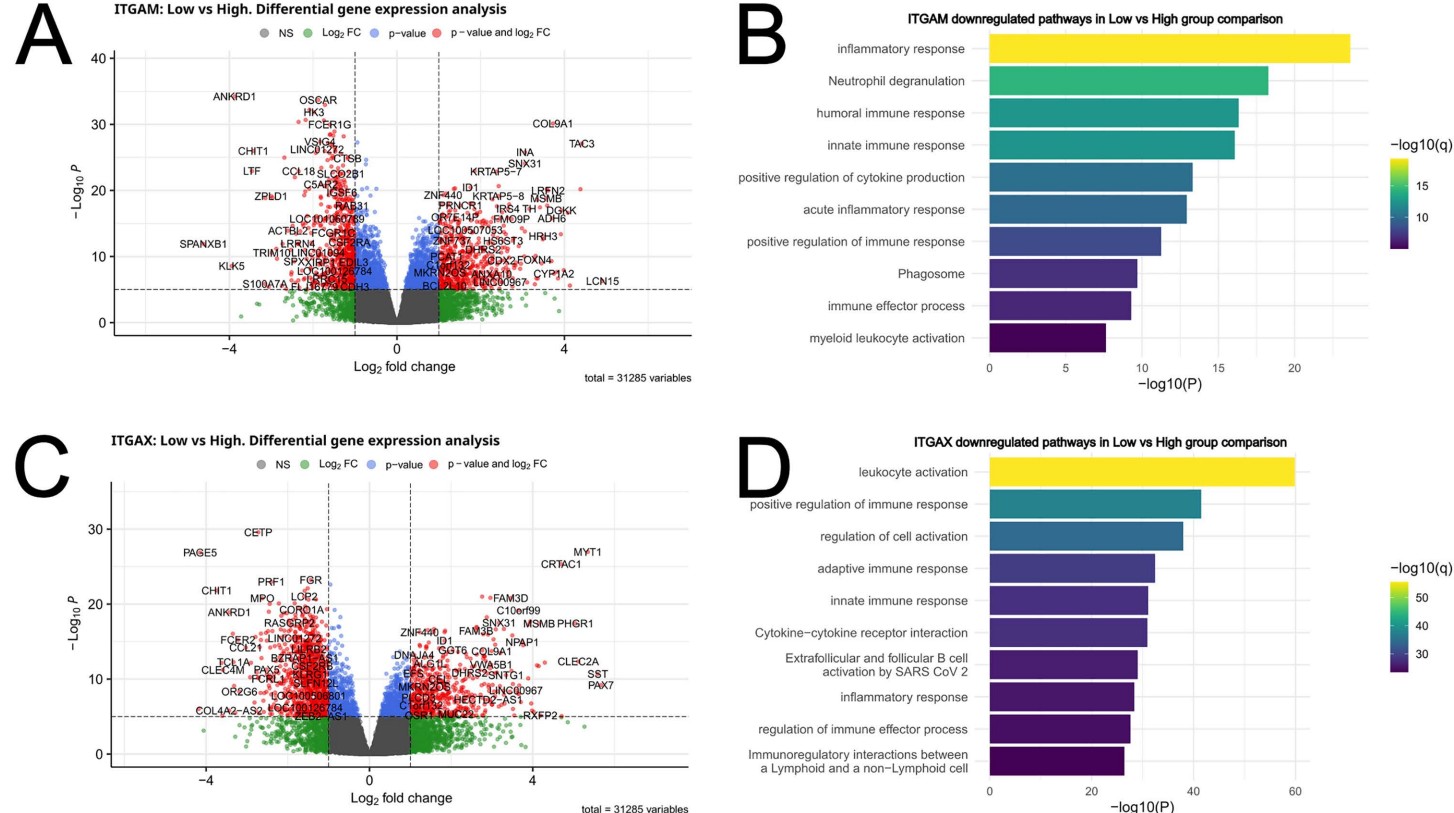

**Fig 5. Differential gene expression analysis and gene set enrichment analysis for *ITGAM* and *ITGAX*.** Volcano plots showing the differentially expressed genes in the group comparison low vs high for *ITGAM* (A) and *ITGAX* **(C)**. All genes that passed the p-value and log2 fold change threshold are indicated in red. Downregulated genes in the comparison have a negative Log2 fold change and upregulated genes have a positive Log2 fold change. Functionally enriched pathways were plotted for *ITGAM* (B) and *ITGAX* **(D)**.

## Association of *FLNA* expression levels with ECM genes

Filamin A is expressed in immune cells but also in other types of cells. Indeed, the differential gene expression analysis of *FLNA* did not reveal an association with immune genes; instead, in *FLNA* low samples pathways such as "NABA core matrisome", "extracellular matrix organization", "response to wounding" and "regulation of cell-substrate adhesion" were affected (Fig 6B). For example, *MYL9* (myosin light chain 9, involved in focal adhesions and integrin-mediated cell adhesion/migration), *ITGA5* (integrin α5), *FLNC* (filamin C, involved in Cell-extracellular matrix interactions and cell-cell communication), *TGFB1I1* (Transforming growth factor β 1 induced transcript 1, involved in cell proliferation and migration) were downregulated in *FLNA* low samples (Fig 6A). Thus, filamin A expression levels seem to regulate mainly the tumor extracellular microenvironment and not directly affect immune genes in the samples.

## Effect of β2-integrins and their regulators on the immune landscape in urothelial cancer

Our differential gene expression analysis indicated differences in the immune landscape in tumors expressing different levels of β2-integrins and their regulators. Next, we used EcoTyper [18] to see if stratified groups based on low and high expression of *ITGAL*, *ITGAM*, and *FLNA* correlated with differences in immune cell types within the tumor environment (Fig 7 and Table 1).

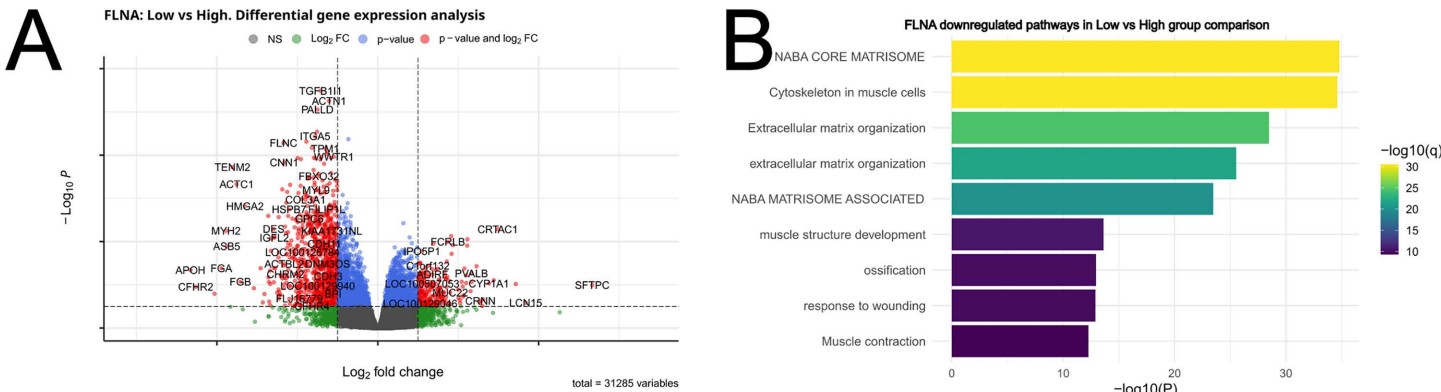

**Fig 6. Differential gene expression analysis and gene set enrichment analysis for *FLNA*.** Volcano plots showing the differentially expressed genes in the group comparison low vs high for *FLNA* **(A)**. All genes that passed the p-value and log2 fold change threshold are indicated in red. Downregulated genes in the comparison have a negative Log2 fold change and upregulated genes have a positive Log2 fold change. Functionally enriched pathways were plotted for *FLNA* **(B)**.

Individuals belonging to the *ITGAL* low group (expressing a low amount of *ITGAL*, Fig 7A) showed lower amounts of Treg (S1, CXCR6+, CTLA4+) and naïve (CCR7+) CD4+ T cells, lower amount of naïve CD8+ T cells (S1, BTLA+, GZMK+) but higher amount of late-stage differentiated effector CD8+ T cells (S2, FCGR3A+, LAIR1+). Notably, the *ITGAL* low group of patients also had a lower amount of classical (PRF1+, CD247+) NK cells. As for myeloid cells, the *ITGAL* low group had a larger number of proliferative (S8, CDK4+), M0 (S2, FABP4+, MARCO+), M1 (CXCL9, SLAMF8+) macrophages, but smaller number of monocytes (S1, CCR2+, CLEC10A+), M2 (CD300E+, CLEC5A+) and M2-like (S1PR1+) macrophages. Dendritic cell amounts also differed, with the *ITGAL* low group having a higher amount of myeloid cDC1 (S1, CLEC9A+, XCR1+), but a lower amount of inflammatory myeloid cDC2-B (S2, *ITGAM*+, CLEC12A+) and mature immunogenic DCs (S3, PRF1+, CD274+, CD80+). *ITGAL* expression therefore significantly impacted on tumor infiltration of especially naïve CD8 + T cells and classical NK cells that have a positive correlation with survival [18]. However, it also affected the myeloid populations in the tumors, which may have a major effect on the response to immunotherapy (Fig 7 and Table 1).

When comparing the *ITGAM* groups (Fig 7B), we identified that the *ITGAM* low group had a higher amount of Treg (S1, CXCR6+, CTLA4+) and resting (KLF2+) CD4+ T cells. As for CD8+ T cells, they had a lower amount of naïve (S1, BTLA+, GZMK+), a bigger number of late-stage differentiated effector (S2, FCGR3A+, LAIR1+), but a similar number of effector memory (S3, IFNG+, GZMB+, LAG3+) CD8+ T cells compared to the *ITGAM* high group. NK cell pools were also different, with *ITGAM* low having less classical (S1, PRF1+, CD247+), but more normal-enriched (S2, STX11+) NK cells. Regarding myeloid cells, both groups had a similar number of monocytes (S1, CCR2+, CLEC10A+), but *ITGAM* low group had a lower amount of M0 (S2, FABPR4+, MARCO+), M2 (S4, CD300E, CLEC5A), and M2-like (S5, S1PR1+) macrophages than the *ITGAM* high group. *ITGAM* low had a larger number of M1 (S3, CXCL9+, SLAMF8+), and proliferative (S8, CDK4+) macrophages than the *ITGAM* high group. There were also differences in the types of dendritic cells, with *ITGAM* low group having a higher amount of myeloid cDC1 (S1, CLEC9A+, XCR1+) but lower amount of inflammatory myeloid cDC2-B (S2, *ITGAM*+, CLEC12A+), mature immunogenic (S3, PRF1+ CD247+, CD80+), mature (S5, CAV1+), and migratory activated (S7, CXCL2+, CXCL8+) dendritic cells. *ITGAM* expression therefore significantly affected especially the myeloid landscape of the tumors (Table 1); In *ITGAM* low samples there were more M1 macrophages and myeloid cDC1 dendritic cells, that show a positive correlation with survival [18].

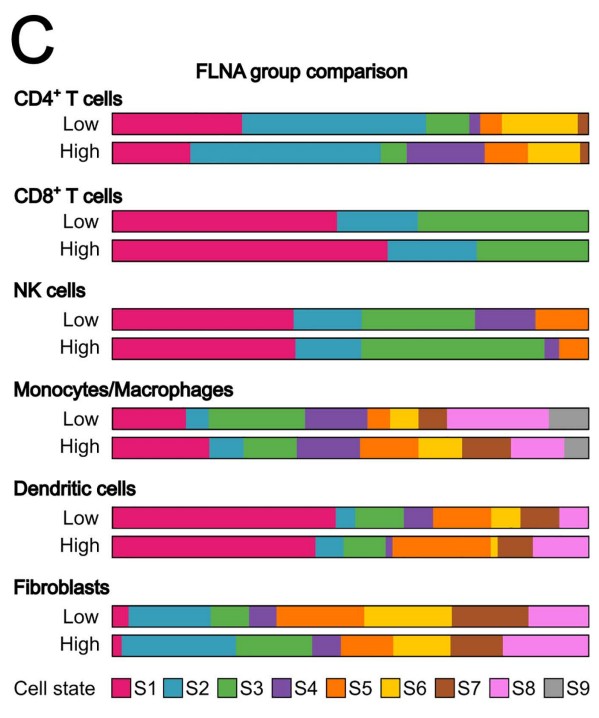

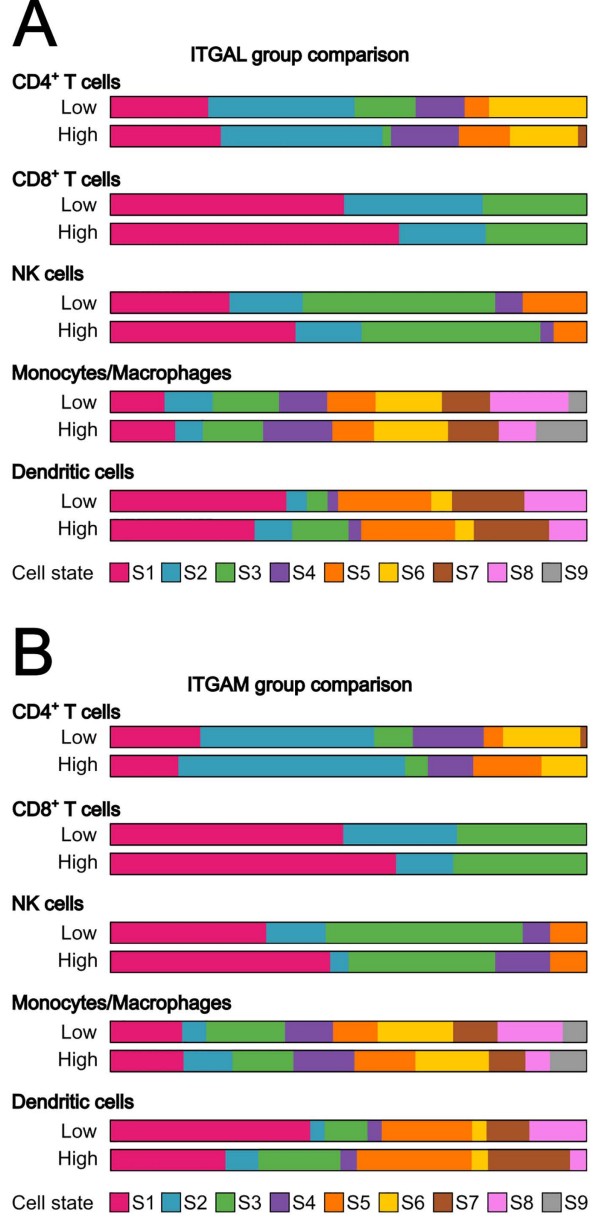

**Fig 7. Comparison of cell states in *ITGAL*, *ITGAM* and *FLNA* groups.** The proportion of different cell states of CD4 + T cells, CD8 + T cells, NK cells, Monocytes/Macrophages and Dendritic cells in the low expressing groups were compared to their corresponding high expressing groups for *ITGAL* **(A)**, *ITGAM* **(B)**, and *FLNA* **(C)**, and are represented with colored bars. Cell states as defined by [18] are shown in the indicated colors.

In the *FLNA* group comparison (Fig 7C), the *FLNA* low group had more Treg (S1, CXCR6+, CTLA4+), slightly less naïve (S2, CCR7+), and substantially less resting (S4, KLF2+) CD4 + T cells than the *FLNA* high group. It also had less naïve (S1, BTLA+, GMZK+) and slightly less late-stage differentiated effector CD8+ T cells (S2, FCGR3A+, LAIR1+) than the *FLNA* high group. There were no big differences among the well characterized NK cell states. In the myeloid cell category, the *FLNA* low group had a lower number of monocytes (S1, CCR2+, CLEC10A+), M0 (S2, FABP4+, MARCO+), and M2-like (S5, S1PR1+), while it had a higher number of M1 (S3, CXCL9+,

**Table 1. Summary of major cell population changes in ITGAL high, ITGAM high and FLNA high tumors detected by EcoTyper analysis. In blue, immunological changes which are thought to be anti-tumorigenic, in red, changes which are thought to be pro-tumorigenic.**

| ITGAL high | ITGAM high | FLNA high |
|---|---|---|
| Treg ↑ | Treg ↓ | Treg ↓ |
| effector CD8 + T cell ↑ | effector CD8 + cell ↓ | naive T cell ↑ |
| NK cell ↑ | naive CD8 + cell ↑ | monocyte ↑ |
| M2 macrophages ↑ | NK cell ↑ | M2 macrophage ↑ |
| M1 macrophages ↓ | M2 macrophage ↑ | M1 macrophage ↓ |
| monocytes ↑ | M1 macrophage ↓ | foamy macrophage ↑ |
| cDC1 ↓ | cDC1 ↓ | activated DC ↓ |
| activated DC ↑ | activated DC ↑ | CAF ↑ |

SLAMF8+), proliferative (S8, CDK4+) than the *FLNA* high group. Importantly, M2 foam cell-like macrophages (S6, AEBP1+), which have been recently shown to be significantly associated with patient survival (18), were decreased in this group. As for dendritic cells, the *FLNA* low group had slightly more myeloid cDC1 (S1, CLEC9A+, XCR1+), mature immunogenic (S3, PRF1+, CD274+, CD80+) and migratory-activated (S7, CXCL2+, CXCL8+), less mature (S5, CAV1+), and slightly less inflammatory myeloid cDC2-B (S2, *ITGAM*+, CLEC12A+) than the *FLNA* high group.

For *FLNA* we also analyzed fibroblasts; *FLNA* low group had especially less CAF2 (S2, CD34+, SPARCL1+), CAF1 (S3, POSTN+, COL10A1+) and pro-migratory-like (S8, CA9+) states, which are significantly associated with lower survival in cancer patients.

In conclusion, *FLNA* expression affects fibroblast phenotype, ECM gene expression and through that, appears to (indirectly) influence also especially myeloid immune landscape (M1/M2 macrophage balance, foamy cell macrophages) in the tumors (Table 1).

## Correlation of β2-integrins and their regulators on carcinoma ecotype in urothelial cancer

We further used EcoTyper to analyze the carcinoma ecotypes [18] in the patient groups expressing different amounts of β2-integrins and their regulators (Fig 8). Carcinoma ecotypes (CEs) represent distinct transcriptional states of tumors that reflect underlying immune microenvironments and are strongly associated with clinical outcomes. These ecotypes serve as robust predictors of patient prognosis and response to immunotherapy, providing a powerful framework for stratifying tumors based on their immune and molecular characteristics. *ITGAM* high group tumors were enriched for CE1/CE2

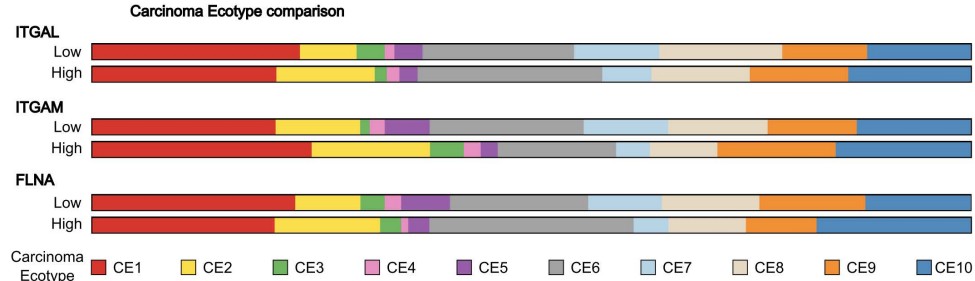

**Fig 8. Carcinoma ecotypes of *ITGAL*, *ITGAM* and *FLNA* groups compared.** The prevalence of the different carcinoma ecotypes in the low and high groups for *ITGAL*, *ITGAM* and *FLNA* was assessed and represented with colored bars. Carcinoma ecotypes as described by [18]are shown in the indicated colors.

carcinoma ecotypes, which are lymphocyte deficient and strongly linked to higher risk of death, in concordance with our findings that the *ITGAM* high group is associated with decreased survival. In addition, *ITGAM* high group tumors were also enriched for the CE3 ecotype, which is myeloid enriched and also associated with decreased survival (Fig 8).

The *ITGAL* high group tumors, which were associated with increased survival in the patients, showed reduced enrichment for the CE1 carcinoma ecotype and modest enrichment for immunogenic CE10 carcinoma ecotype, which is linked to naïve/central memory T cells, increased naïve B cell content and cDC1 dendritic cells. However, *ITGAL* high tumors did not show an enrichment of the CE9 ecotype (characterized by IFNγ signaling and T cell exhaustion), which is the best predictor of responsiveness to immunotherapy (Fig 8). Interestingly, the *FLNA* low group, which did display an association with increased responsiveness to immunotherapy, did have an enrichment of CE9 carcinoma ecotype, showing that filamin expression, either directly or indirectly, does influence the immunological landscape of urothelial carcinoma (Fig 8).

## Discussion

In this paper, we have conducted a series of bioinformatic analyses to investigate the role of the different β2-integrins in urothelial cancer and responsiveness to immunotherapy. This study reveals fundamentally different roles of different β2-integrins in tumor development and in determining the immune landscape of the tumors. Of note, the integrin genes discussed here are immune cell specific and will therefore not be found on, e.g., circulating cancer cells. *ITGAL* and *FERMT3*, encoding for the αL chain of LFA-1 and for kindlin-3, respectively, are positively associated with patient survival. Interestingly, ITGAL has recently been implicated in a pan-cancer study as a cancer biomarker correlating with increased survival [19].

Our gene expression and cancer Ecotyper analyses reveal that they appear to be most important in regulating adaptive (e.g., T cell mediated) responses to the tumors, but surprisingly, their expression level does not correlate with responsiveness to immunotherapy (although T cell infiltration is thought to be one of the most important determinants of anti-tumor immunity, and B cell responses have also recently been implicated [20]). In other studies, ITGAL expression has been correlated with immune infiltration and good treatment response to immunotherapy in HNSCC and NSCLC [19,21]. However, T cell infiltration per se does not always correlate with better response to immunotherapy, as this is also highly dependent on T cell subtype. Tregs are immunosuppressive T cells that express high levels of *ITGAL*, which is also important for Treg suppressive function [22]. The *ITGAL* low samples had lower levels of Tregs than *ITGAL* high samples. Consequently, larger amounts of Tregs in *ITGAL* high tumors may contribute to a highly immunosuppressive tumor environment, which may not be possible to combat with check-point inhibitor treatment. Another reason might be that *ITGAL* and *FERMT3* expression may not correlate with T cell state, e.g., exhaustion, which is being targeted with immunotherapy approaches. In addition, an adaptive anti-tumor immune response *per se* may not be enough to ensure a good treatment response to immunotherapy if pro-tumorigenic myeloid inflammation is dominant. Indeed, the balance between these two has been shown to be predictive of treatment response to immunotherapy [23]. It is worthwhile to note that *ITGAL* and *FERMT3* are not only expressed in cells of the adaptive immune system, but also in myeloid cells (although they are not myeloid markers *per se)*.

In contrast to *ITGAL* and *FERMT3*, myeloid genes, e.g., *ITGAM* and *ITGAX*, show a negative correlation with survival, and regulate both the myeloid immune landscape of the tumors as well as their responsiveness to immunotherapy. *ITGAX* has also previously been correlated with poor survival in other cancers (gastric cancer) [24]. Our findings here are in good agreement with recent studies associating inflammatory myeloid phagocytic cells with poor treatment responses to immunotherapy in urothelial cancer [25,23].

Interestingly, *FLNA*, encoding for filamin A, an important regulator of integrins and cell adhesion, also regulated the immunological landscape of tumors, and its expression is inversely correlated with patient survival and response to immunotherapy. As filamin A is expressed both in immune cells and in other cell types, its effects on the immunological

landscape may not be direct. Indeed, our gene expression and EcoTyper analyses suggest that filamin A expression affects mainly fibroblast phenotype and the expression of extracellular matrix proteins, which may indirectly affect the immune profile of the tumors. The tumor microenvironment, including cancer associated fibroblasts, plays vital roles in shaping tumor progression and immune response in urothelial cancer. A highly crosslinked ECM, produced by CAFs, may function as a physical barrier to immune (T cell) infiltration. There is also extensive crosstalk between CAFs and tumor associated macrophages, contributing further to an immune suppressive environment of tumors [26]. Lack of response to immunotherapy in the Imvigor cohort has previously been shown to correlate with TGF-beta signaling in fibroblasts, which affects deep infiltration of cytotoxic T cells into the tumor parenchyma [27]. Interestingly, filamin A has been shown to be important for TGF-beta signaling, by affecting downstream Smad proteins [28]. Thus, *FLNA* expression may correlate with TGF-beta signalling and thus affect the immune suppressive environment of tumors, and therefore the response to immunotherapy.

An obvious limitation of this study is that it uses only one dataset, and therefore sample size was limited. In the future, further research in multiple datasets and tumor types is required to confirm the results and may expand the findings also to other cancers and immunotherapy responses.

In conclusion, this study reveals that expression of specific β2-integrin subunits and regulators particularly *ITGAL*, *FERMT3*, *ITGAM* and *FLNA* may be explored in the future as biomarkers to differentiate urothelial cancer patients with different immune landscapes, survival profiles and responsiveness to immunotherapy.

## Methods

### Urothelial cancer data

Transcriptome-wide gene expression data and survival data was obtained from IMvigor210 urothelial cancer cohort, available via the R package IMvigor210CoreBiologies, license under the Creative Commons 3.0 and available for free download at http://research-pub.gene.com/IMvigor210CoreBiologies/. To ensure data completeness for survival analyses, 50 individuals lacking survival data information were excluded, resulting in 298 patients used for downstream analysis.

### Categorization of patients

To evaluate the association between gene expression levels and patient survival, the 298 patients from IMvigor210 cohort were categorized into "Low" and "High" expression groups for each gene of interest *ITGB2*, *ITGAL*, *ITGAM*, *ITGAX*, *ITGAD*, *FERMT3*, *TLN1* and *FLNA*. This was done using the surv_cutpoint() function from the "survminer" R package (https://rpkgs.datanovia.com/survminer/index.html). The categorization employed log-rank statistics and accounted for patient survival, as done in [29–31]. Thus, the continuous variable of gene expression for each of the genes of interest was transformed into a discrete variable and optimal cutoff values determined for each of the genes (Supplementary S1 Table).

### Survival analysis

Survival analysis was done using the function survfit from the R package "survival" (https://cran.r-project.org/package=survival). Kaplan-Meier estimates of survival curves were plotted using the R package "survminer". Statistical significance between "Low" and "High" expression level groups of each of the eight genes of interest was determined using the log-rank (Mantel-Cox) test.

### Clinical response rate to anti-PD-L1 immunotherapy

Following the categorization of patients into "Low" and "High" expression level groups for each of the eight genes of interest, groups were compared based on their clinical response to immunotherapy (SD/PD, stable disease/

progressive disease, CR/PR, complete response/partial response) and plotted. Statistical analysis was done using the chi-squared test.

## Differential gene expression analysis

Differential gene expression analysis (DGEA) was performed using "DESeq2" R package (http://www.bioconductor.org/packages/release/bioc/html/DESeq2.html). A total of eight DGEA were performed (one for each gene of interest). Each of the DGEA was performed using a count matrix that excluded the gene of interest from its own analysis to avoid bias in the results (e.g., ITGAM was excluded from the count matrix of the DGEA relative to ITGAM expression). The differentially expressed genes (DEGs) were identified by comparing "Low" group vs "High" group for each of the genes of interest, at adjusted p-values < 0.05. The results were visualized using volcano plots using EnhancedVolcano R package (https://github.com/kevinblighe/EnhancedVolcano).

## Functional enrichment analysis

Functional enrichment analysis was performed to identify key biological processes and pathways linked to differentially expressed genes (DEGs) between the Low vs High conditions. Gene Ontology (GO) and Kyoto Encyclopedia of Genes and Genomes (KEGG) pathway analyses were mainly carried out using Metascape [32], which uses consolidated multiple up-to-date biological databases. DEGs were filtered based on an absolute log2 fold change threshold of |1.5| and a false discovery rate (FDR)-adjusted p-value ≤ 0.05, and only those meeting these criteria were included in the enrichment analysis.

## EcoTyper analysis

Gene expression data of the urothelial cohort standardized as transcript per million (TPM) normalized to gene length and sequencing depth, and categorized into groups (low/high) for *ITGAL*, *ITGAM*, and *FLNA* was inputted to EcoTyper (https://ecotyper.stanford.edu/).

## Supporting information

**S1 Table. Cutoff values for each of the studied genes.**
(XLSX)

## Author contributions

**Conceptualization:** Susanna C Fagerholm.

**Formal analysis:** Marc Llort Asens, Imran Khan.

**Funding acquisition:** Susanna C Fagerholm.

**Investigation:** Susanna C Fagerholm, Marc Llort Asens.

**Methodology:** Marc Llort Asens, Imran Khan.

**Project administration:** Susanna C Fagerholm.

**Resources:** Susanna C Fagerholm.

**Supervision:** Susanna C Fagerholm.

**Visualization:** Marc Llort Asens.

**Writing – original draft:** Susanna C Fagerholm, Marc Llort Asens.

**Writing – review & editing:** Susanna C Fagerholm, Marc Llort Asens, Imran Khan.

## Acknowledgments

We would like to thank Imrul Faisal and Daniel Davies for insightful discussions.

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
