## [Decision Letter · Decision Letter 0]

12 Aug 2025

Dear Dr. Fagerholm,

Thank you for submitting your manuscript to PLOS ONE. After careful consideration, we feel that it has merit but does not fully meet PLOS ONE’s publication criteria as it currently stands. Therefore, we invite you to submit a revised version of the manuscript that addresses the points raised during the review process.

We look forward to receiving your revised manuscript.

Kind regards,

Xing-Xiong An, M.D.

Academic Editor

PLOS ONE

Journal Requirements:

[Research Council of Finland

Liv och Hälsa foundation

Magnus Ehrnrooth Foundation]. 

[We would like to thank Imrul Faisal and Daniel Davies for insightful discussions. This study was funded by Research council of Finland, Magnus Ehrnrooth foundation and Liv och Hälsa foundation (all to S.C.F.).]

[Research Council of Finland

Liv och Hälsa foundation

Magnus Ehrnrooth Foundation]

Reviewers' comments:

Reviewer's Responses to Questions

**Comments to the Author**

1. Is the manuscript technically sound, and do the data support the conclusions?

Reviewer #1: Yes

Reviewer #2: Yes

Reviewer #3: Yes

Reviewer #4: Yes

Reviewer #5: Partly

2. Has the statistical analysis been performed appropriately and rigorously?

Reviewer #1: Yes

Reviewer #2: Yes

Reviewer #3: I Don't Know

Reviewer #4: Yes

Reviewer #5: Yes

3. Have the authors made all data underlying the findings in their manuscript fully available?

Reviewer #1: Yes

Reviewer #2: Yes

Reviewer #3: Yes

Reviewer #4: Yes

Reviewer #5: Yes

4. Is the manuscript presented in an intelligible fashion and written in standard English?

Reviewer #1: Yes

Reviewer #2: Yes

Reviewer #3: Yes

Reviewer #4: Yes

Reviewer #5: Yes

Reviewer #1: -choose a different short title than the title itself.

- Write the name of all genes in italics.

-Kindly add more details on how gene expression was performed.

- Elaborate more in the discussion as it was a summary of the results and not a discussion.

- Go over all punctuation errors and few typo errors.

Reviewer #2: This study presents a comprehensive bioinformatic investigation of β2-integrins and their cytoplasmic regulators in urothelial cancer using the IMvigor210 cohort. The work is timely and addresses an underexplored aspect of tumor immunobiology, proposing that distinct β2-integrin subunits are differentially associated with patient prognosis and response to immunotherapy. The idea of using β2-integrins as biomarkers is novel and clinically relevant, given the current limitations in predicting immunotherapy outcomes.

* All findings are derived from a single public dataset. The study would benefit from validation in an independent urothelial cancer cohort (e.g., TCGA BLCA or in-house data), particularly for biomarker claims. If this is not possibile for this project please list it in the limitation section and propose it as future project. This should be linked to mild conclusion.

* Line 32-35: references is needed. Please consider doi: 10.3390/cancers16061115.

* Discussion: The functional interpretation of some associations, especially regarding FLNA, remains speculative. Further discussion is needed to clarify whether its link to ECM remodeling could causally impact immune exclusion or therapy resistance.

* It is unclear how TPM values were normalized prior to EcoTyper input. This should be specified, given that improper normalization may skew cell state predictions. Please clarify.

* Some pathway analyses are described in a repetitive way and could be streamlined (e.g., gene set enrichment for both ITGAL and FERMT3.)

Reviewer #3: This article has been written and well and the topic is novel. I congratulate the authors on this paper.

There are no major flaws with this paper.

The grammar is good

The discussion can be more concise to the reader as below:

As a treating Urologist the authors can sum up in a table the integrins which can be used best in clinical practice, cost of the test, which tissue can be used (blood or urine or cancer tissue), , clinical implications of that integrins test.

It will also be worthwhile to discuss if these integrins can be detected in circulating tumour cells picked up on these urothelial cancer patients.

Reviewer #4: The manuscript by Asens et al. presents a comprehensive transcriptomic analysis of β2-integrin family members and their cytoplasmic regulators in urothelial cancer using the IMvigor210 dataset. The authors demonstrate that high expression of ITGAL and FERMT3 correlates with improved overall survival, whereas high expression of ITGAM, ITGAX, and FLNA is associated with poorer survival and reduced response to anti–PD-L1 immunotherapy. The study is well-structured and integrates multiple layers of analysis—including survival, immunotherapy response, differential gene expression, pathway enrichment, and EcoTyper-based immune cell deconvolution and ecotype classification. However, certain conclusions are stated too strongly given the correlational nature of the data, and limitations related to subgroup sizes and the absence of external validation should be more clearly acknowledged.

Specific Comments

1. The observation that ITGAL and FERMT3 are linked to better survival but not to improved immunotherapy response is interesting and merits deeper discussion. Consider addressing potential roles of T cell exhaustion or the possibility that T cell infiltration does not always imply functional anti-tumor activity.

2. The findings rely entirely on the IMvigor210 dataset. Validation using an independent cohort (e.g., TCGA-BLCA or other publicly available datasets) would strengthen the conclusions. If not feasible, this limitation should be stated.

3. Figure 7 contains valuable information, but the abundance of details may overwhelm readers. A table or simplified schematic summarizing key differences in immune cell populations between high and low expression groups would enhance clarity.

Reviewer #5: This is an interesting paper that for the first time explores the prognostic significance of beta2 integrins in the IMvigor210 cohort of patients treated with atezolizumab for advanced or metastatic urothelial cancer. At a high level, the study confirms previous findings regarding the relationships between inferred T-cell and myeloid populations and survival outcomes. The study must be considered exploratory given that it relied exclusively on bulk RNAseq data from a single patient cohort; as detailed below, the same biomarkers should be evaluated in additional public cohorts. Detailed comments follow.

1. The prognostic relationships between beta2 integrin mRNA expression and survival outcomes should also be explored in datasets from patients who were not treated with an immune checkpoint inhibitor. Recent data from the Alliance 90601 Phase-3 clinical trial (performed in the same patient population) and from TCGA (performed in a different disease state) would both be appropriate.

2. The relationships should also be explored in additional bulk RNAseq datasets from patients treated with immune checkpoint blockade.

3. The inferred relationships between beta2 integrin expression and other genes

should be confirmed in public bladder cancer scRNAseq datasets.

4. A recent study explored the role of B-cell gene expression in the same cohort (PMID: 34426176). How did the beta2 integrin expression levels correlate with B-cell genes?

5. Other recent studies, particularly from Galsky and colleagues, implicated myeloid cell populations immune checkpoint blockade resistance and poor outcomes in urothelial cancer patients. These other studies should be cited and discussed.

6. The connection between FLNA and fibroblasts should also be explored further, preferably using scRNAseq data. Past studies implicating TGFbeta and fibroblasts in resistance and/or shorter survival the same cohort and others should be discussed.

**Do you want your identity to be public for this peer review?** For information about this choice, including consent withdrawal, please see our Privacy Policy

Reviewer #1: No

Reviewer #2: No

Reviewer #3: **Yes: ** Danny Darlington Carbin

Reviewer #4: No

Reviewer #5: **Yes: ** David J. McConkey

---

## [Author Response · Author response to Decision Letter 1]

15 Sep 2025

Response to reviewers

Please find below the reviewer comments, and our responses to these.

Reviewer #1:

-choose a different short title than the title itself.

We have changed the short title, as requested.

- Write the name of all genes in italics.

We apologize for this oversight. The gene names have now been changed to italics.

-Kindly add more details on how gene expression was performed.

We have tried to elaborate on the details, as requested, we hope this is now adequately described.

- Elaborate more in the discussion as it was a summary of the results and not a discussion.

We have now elaborated on the Discussion, as requested. We discuss in more detail reasons why ITGAL and FERMT3 expression level does not correlate with immunotherapy responses (for example, Tregs may play a role here, which express high levels of both proteins; the balance between pro-tumorigenic myeloid inflammatory responses and anti-tumorigenic adaptive responses is also discussed). We also further discuss FLNA, fibroblasts and TGFbeta pathway in treatment response.

- Go over all punctuation errors and few typo errors.

We have tried to correct these, thank you for pointing this out.

Reviewer #2: This study presents a comprehensive bioinformatic investigation of β2-integrins and their cytoplasmic regulators in urothelial cancer using the IMvigor210 cohort. The work is timely and addresses an underexplored aspect of tumor immunobiology, proposing that distinct β2-integrin subunits are differentially associated with patient prognosis and response to immunotherapy. The idea of using β2-integrins as biomarkers is novel and clinically relevant, given the current limitations in predicting immunotherapy outcomes.

* All findings are derived from a single public dataset. The study would benefit from validation in an independent urothelial cancer cohort (e.g., TCGA BLCA or in-house data), particularly for biomarker claims. If this is not possibile for this project please list it in the limitation section and propose it as future project. This should be linked to mild conclusion.

We fully agree with the reviewer that this is a limitation of our study, and that more studies are required in the future to confirm our findings here. We have added text to this effect in the discussion, and emphasize that further studies are needed before we can make conclusions about the use of these genes as biomarkers in urothelial cancer.

* Line 32-35: references is needed. Please consider doi: 10.3390/cancers16061115.

Thank you for this suggestion, we have inserted this reference.

* Discussion: The functional interpretation of some associations, especially regarding FLNA, remains speculative. Further discussion is needed to clarify whether its link to ECM remodeling could causally impact immune exclusion or therapy resistance.

We have expanded on the Discussion section to clarify these links, particularly with regards to FLNA, and its putative links to TGF-beta signaling.

* It is unclear how TPM values were normalized prior to EcoTyper input. This should be specified, given that improper normalization may skew cell state predictions. Please clarify.

We thank the reviewer for this important point. We have now specified that the TPM normalization was done taking gene length and sequencing depth into account.

* Some pathway analyses are described in a repetitive way and could be streamlined (e.g., gene set enrichment for both ITGAL and FERMT3.)

We appreciate the reviewer’s comment and we have tried to further streamline sections of the paper (e.g. by creating Table 1, which summarizes the main findings from Figure 7).

Reviewer #3: This article has been written and well and the topic is novel. I congratulate the authors on this paper.

There are no major flaws with this paper.

The grammar is good

We thank the reviewer for these positive comments on our study.

The discussion can be more concise to the reader as below:

As a treating Urologist the authors can sum up in a table the integrins which can be used best in clinical practice, cost of the test, which tissue can be used (blood or urine or cancer tissue), , clinical implications of that integrins test.

Although we understand the reviewer’s point, the authors are not clinicians, and therefore we are not specialists on clinical tests or implications for these. We therefore do not feel qualified to insert this kind of data into the paper. We also emphasize in the paper that more studies are needed to draw definitive conclusions on these biomarkers; this is just a first study.

It will also be worthwhile to discuss if these integrins can be detected in circulating tumour cells picked up on these urothelial cancer patients.

This is a good point. The integrins we discuss here are leukocyte specific (ITGB2, ITGAL, ITGAM, ITGAX), so will not be found on cancer cells. We now mention this in the discussion.

Reviewer #4: The manuscript by Asens et al. presents a comprehensive transcriptomic analysis of β2-integrin family members and their cytoplasmic regulators in urothelial cancer using the IMvigor210 dataset. The authors demonstrate that high expression of ITGAL and FERMT3 correlates with improved overall survival, whereas high expression of ITGAM, ITGAX, and FLNA is associated with poorer survival and reduced response to anti–PD-L1 immunotherapy. The study is well-structured and integrates multiple layers of analysis—including survival, immunotherapy response, differential gene expression, pathway enrichment, and EcoTyper-based immune cell deconvolution and ecotype classification. However, certain conclusions are stated too strongly given the correlational nature of the data, and limitations related to subgroup sizes and the absence of external validation should be more clearly acknowledged.

Specific Comments

1. The observation that ITGAL and FERMT3 are linked to better survival but not to improved immunotherapy response is interesting and merits deeper discussion. Consider addressing potential roles of T cell exhaustion or the possibility that T cell infiltration does not always imply functional anti-tumor activity.

This is a good point, we thank the reviewer for this suggestion. We have now further commented on these possibilities in the Discussion section, eg for example Treg contribution to the immunosuppressive microenvironment in the ITGAL high tumors. We further discuss the balance between adaptive anti-tumor responses and pro-tumorigenic myeloid inflammation, which appears to be a major determinant for treatment response to immunotherapy in urothelial cancer (doi: 10.1158/1078-0432.CCR-20-4574)

2. The findings rely entirely on the IMvigor210 dataset. Validation using an independent cohort (e.g., TCGA-BLCA or other publicly available datasets) would strengthen the conclusions. If not feasible, this limitation should be stated.

We fully agree that this is a limitation of our study. We now clearly state this limitation in the article, as requested, and state that further work in independent cohorts is required to confirm these results before firm conclusions can be drawn.

3. Figure 7 contains valuable information, but the abundance of details may overwhelm readers. A table or simplified schematic summarizing key differences in immune cell populations between high and low expression groups would enhance clarity.

We agree and we have now inserted a Table with the key differences, which we hope is helpful for clarity.

ITGAL high ITGAM high FLNA high

Treg ↑ Treg ↓ Treg ↓

effector CD8+ T cell ↑ effector CD8+ cell ↓ naive T cell ↑

NK cell ↑ naive CD8+ cell ↑ monocyte ↑

M2 macrophages ↑ NK cell ↑ M2 macrophage ↑

M1 macrophages ↓ M2 macrophage ↑ M1 macrophage ↓

monocytes ↑ M1 macrophage ↓ foamy macrophage ↑

cDC1 ↓ cDC1 ↓ activated DC ↓

activated DC ↑ activated DC ↑ CAF ↑

Table 1. Summary of major cell population changes in ITGAL high, ITGAM high and FLNA high tumors detected by EcoTyper analysis. In blue, immunological changes which are thought to be anti-tumorigenic, in red, changes which are thought to be pro-tumorigenic.

Reviewer #5: This is an interesting paper that for the first time explores the prognostic significance of beta2 integrins in the IMvigor210 cohort of patients treated with atezolizumab for advanced or metastatic urothelial cancer. At a high level, the study confirms previous findings regarding the relationships between inferred T-cell and myeloid populations and survival outcomes. The study must be considered exploratory given that it relied exclusively on bulk RNAseq data from a single patient cohort; as detailed below, the same biomarkers should be evaluated in additional public cohorts. Detailed comments follow.

We thank the reviewer for their valuable and very insightful comments on our study, they have been very helpful to improve the manuscript. We fully agree that further studies are needed in other cohorts. However, we think this is beyond the scope of the current paper and should instead be included in future investigations, which we emphasize should be performed before firm conclusions can be drawn. In this paper, we have instead in depth analyzed the different beta2-integrins and their regulators in one urothelial cancer cohort, correlation with survival and response to immunotherapy, as well as their role in regulating the immune landscape and immune phenotype of the tumors. This is already a significant body of work, with multiple analyses, which we believe can lay the foundation for work done in other cohorts in the future.

1. The prognostic relationships between beta2 integrin mRNA expression and survival outcomes should also be explored in datasets from patients who were not treated with an immune checkpoint inhibitor. Recent data from the Alliance 90601 Phase-3 clinical trial (performed in the same patient population) and from TCGA (performed in a different disease state) would both be appropriate.

Of the genes investigated here, interestingly, ITGAL has recently been implicated in a pan-cancer study as a cancer biomarker correlating with increased survival (doi: 10.3389/fphar.2024.1464830.) In this study, like in our study, it was reported that ITGAL expression correlates with immune infiltration. In contrast, Itgb2 has been correlated with poor prognosis in glioma (doi: 10.1007/s00262-021-03022-2), and ITGAX with poor survival in gastric cancer (doi: 10.3389/fphar.2024.1536478). Therefore, there is support from the literature for these genes as biomarkers in different cancers. However, we completely agree with the reviewer that additional analyses are required to confirm the results reported here, as these results are based on one study. We mention this limitation of the current study in the Discussion section, and emphasize that further analyses in additional cohorts are required before firm conclusions can be made.

2. The relationships should also be explored in additional bulk RNAseq datasets from patients treated with immune checkpoint blockade.

ITGAL has recently been reported to be linked to sensitivity to immune checkpoint blockade in HNSCC, and NSCLC, which we did not find here. doi: 10.3389/fphar.2024.1464830, doi: 10.3389/fimmu.2024.1382231. We mention these studies in the Discussion section of the paper. Furthermore, we now discuss what the reasons might be for ITGAL not being linked to better immunotherapy response in this study.

3. The inferred relationships between beta2 integrin expression and other genes

should be confirmed in public bladder cancer scRNAseq datasets.

We thank the reviewer for this valuable suggestion. We of course agree that further studies are needed in other cohorts. However, we think this is beyond the scope of the current paper and should instead be included in future investigations, which we emphasize should be performed before firm conclusions can be drawn.

4. A recent study explored the role of B-cell gene expression in the same cohort (PMID: 34426176). How did the beta2 integrin expression levels correlate with B-cell genes?

We indeed found B cell genes to be enriched in ITGAL samples (Fig 4B). ITGAL is highly expressed both in B cells and T cells and is involved also in B cell recruitment. We now mention in the Discussion that B cell responses may also play a role.

5. Other recent studies, particularly from Galsky and colleagues, implicated myeloid cell populations immune checkpoint blockade resistance and poor outcomes in urothelial cancer patients. These other studies should be cited and discussed.

We thank the reviewer for this very useful suggestion. We now cite these studies and discuss them in the paper, as requested. We think they provide a very good framework for the understanding of the results from this study.

6. The connection between FLNA and fibroblasts should also be explored further, preferably using scRNAseq data. Past studies implicating TGFbeta and fibroblasts in resistance and/or shorter survival the same cohort and others should be discussed.

We thank the reviewer for these insights, which we found very useful. We now discuss FLNA, TGFbeta and fibroblast in light of previous findings in the same cohort (doi: 10.1038/nature25501), as requested.

---

## [Decision Letter · Decision Letter 1]

28 Sep 2025

β2-integrins as biomarkers in urothelial cancer

PONE-D-25-26951R1

Dear Dr. Fagerholm,

We’re pleased to inform you that your manuscript has been judged scientifically suitable for publication and will be formally accepted for publication once it meets all outstanding technical requirements.

Kind regards,

Xing-Xiong An, M.D.

Academic Editor

PLOS ONE

Additional Editor Comments (optional):

Reviewers' comments:

Reviewer's Responses to Questions

**Comments to the Author**

Reviewer #2: All comments have been addressed

Reviewer #3: All comments have been addressed

Reviewer #4: All comments have been addressed

2. Is the manuscript technically sound, and do the data support the conclusions?

Reviewer #2: Yes

Reviewer #3: Yes

Reviewer #4: Yes

3. Has the statistical analysis been performed appropriately and rigorously?

Reviewer #2: Yes

Reviewer #3: Yes

Reviewer #4: I Don't Know

4. Have the authors made all data underlying the findings in their manuscript fully available?

Reviewer #2: Yes

Reviewer #3: Yes

Reviewer #4: Yes

5. Is the manuscript presented in an intelligible fashion and written in standard English?

Reviewer #2: Yes

Reviewer #3: Yes

Reviewer #4: Yes

Reviewer #2: The manuscript has improved significantly, addressing previous concerns with clarity and depth. I am satisfied with the improvements made.

Reviewer #3: The authors have addressed my comments.

The introduction methods section is good.

The discussion includes salient studies and topics

The results section had been well presented.

The conclusion is based on the study findings.

Reviewer #4: (No Response)

**Do you want your identity to be public for this peer review?** For information about this choice, including consent withdrawal, please see our Privacy Policy

Reviewer #2: **Yes: ** Savio D Pandolfo

Reviewer #3: **Yes: ** Danny Darlington Carbin

Reviewer #4: No

---

## [Editor Report · Acceptance letter]

PONE-D-25-26951R1

PLOS ONE

Dear Dr. Fagerholm,

I'm pleased to inform you that your manuscript has been deemed suitable for publication in PLOS ONE. Congratulations! Your manuscript is now being handed over to our production team.

Kind regards,

on behalf of

Dr. Xing-Xiong An

Academic Editor

PLOS ONE